# Significant enhancement of proton conductivity in solid acid at the monolayer limit

Zhangcai Zhang[1,2,4], Lixin Liang[3,4], Jianze Feng[1], Guangjin Hou [3] & Wencai Ren [1,2] ✉

Proton transport in nanofluidic channels is not only fundamentally important but also essential for energy applications. Although various strategies have been developed to improve the concentration of active protons in the nanochannels, it remains challenging to achieve a proton conductivity higher than that of Nafion, the benchmark for proton conductors. Here, taking $H_3Sb_3P_2O_{14}$ and $HSbP_2O_8$ as examples, we show that the interactions between protons and the layer frameworks in layered solid acid $H_nM_nZ_2O_{3n+5}$ are substantially reduced at the monolayer limit, which significantly increases the number of active protons and consequently improves the proton conductivities by $\sim$8 – 66 times depending on the humidity. The membranes assembled by monolayer $H_3Sb_3P_2O_{14}$ and $HSbP_2O_8$ nanosheets exhibit in-plane proton conductivities of ~ 1.02 and 1.18 S cm$^{-1}$ at 100% relative humidity and 90 °C, respectively, which are over 5 times higher than the conductivity of Nafion. This work provides a general strategy for facilitating proton transport, which will have broad implications in advancing both nanofluidic research and device applications from energy storage and conversion to neuromorphic computing.

Constructing nanofluidic channels for proton transport is not only fundamentally important for exploring the unusual nanofluidic phenomena[1–5] but also essential for applications in energy storage and conversion, such as fuel cells, batteries, and supercapacitors[6–9]. The fast proton transfer in aqueous systems is commonly governed by the Grotthuss mechanism, which takes place on a hydrogen-bonded network and entails a collective proton motion similar to Newton's cradle[10–13]. The efficiency of Grotthuss-type proton conduction mainly relies on the proton bridges interconnected by the hydrogen bonds, and short hydrogen bonds would generate a superharmonic behavior of proton motion, triggering proton transfer in a nearly barrierless manner[13,14]. Thus, improving the concentration of active protons in the nanochannels to shorten the hydrogen bonds is vital to achieve a high proton conductivity.

Nafion (DuPont Company), the benchmark for proton conductor, is composed of a perfluorinated polymer skeleton and side chains containing sulfonic acid groups ($-SO_3H$) as proton donors[15,16]. The low content of $-SO_3H$ in the nanochannels is a hurdle to get a high proton concentration, resulting in a conductivity up to $\sim$0.2 S cm$^{-1}$ at high relative humidity (RH) and temperature[16,17]. Metal-organic frameworks (MOFs) and covalent-organic frameworks (COFs) are two emerging porous materials for proton conductors thanks to their regular nanochannels[18–21]. Although various strategies, such as frameworks' functionalization and guest molecule inclusion, have been developed to improve the concentration of protons in the nanochannels, it is usually difficult to achieve a proton conductivity higher than that of Nafion[20,21]. Recently, negatively charged vacancies have been found to be an efficient proton donor to improve the proton conductivity of 2D

[1]Shenyang National Laboratory for Materials Science, Institute of Metal Research, Chinese Academy of Sciences, 72 Wenhua Road, Shenyang 110016, China. [2]School of Materials Science and Engineering, University of Science and Technology of China, 72 Wenhua Road, Shenyang 110016, China. [3]State Key Laboratory of Catalysis, Dalian National Laboratory for Clean Energy, Dalian Institute of Chemical Physics, Chinese Academy of Sciences, Dalian 116023, China. [4]These authors contributed equally: Zhangcai Zhang, Lixin Liang. ✉e-mail: wcren@imr.ac.cn

materials assembled membranes[22]. However, such membranes are unstable in an acidic environment, and thus it remains a challenge to obtain pure proton conductors.

Solid acids $H_nM_nZ_2O_{3n+5}$ (M = Sb, Nb, Ta; Z = P, As; n = 1, 3) are a large family of layer-structured proton conductors with abundant protons lying in $M_nZ_2O^{n-}{}_{3n+5}$ interlayers[23–26]. Taking $H_nSb_nP_2O_{3n+5}$ as an example, there are two kinds of protons trapped on the bridging oxygen atoms of P-O-Sb and Sb-O-Sb[24], respectively, for $H_3Sb_3P_2O_{14}$ (n = 3) (Fig. 1a–c), while only one kind of protons are trapped on the bridging oxygen atoms of P-O-Sb for $HSbP_2O_8$ (n = 1)[25]. Bulk $H_nSb_nP_2O_{3n+5}$ has been used previously as proton conductors by pressing $H_nSb_nP_2O_{3n+5}$ particles together into a pellet because big enough crystals were not generally possible to obtain[26]. However, the weak binding between $H_nSb_nP_2O_{3n+5}$ particles limits their use to RH lower than 95% and temperature below 20 °C, beyond which the pellet suffers severe expansion and cracking. The reported proton conductivities of bulk $H_nSb_nP_2O_{3n+5}$ are typically in the range of $10^{-5}$ to $10^{-2}$ S cm$^{-1}$ from 20 to 95% RH at 20 °C[26].

Here, we show that the interactions between protons and the layer frameworks in $H_nSb_nP_2O_{3n+5}$ are substantially reduced at the monolayer limit, which significantly increases the number of active protons and consequently improves the proton conductivities by ~8–66 times depending on the humidity. The membranes assembled by monolayer $H_3Sb_3P_2O_{14}$ (m-$H_3Sb_3P_2O_{14}$) and $HSbP_2O_8$ (m-$HSbP_2O_8$) nanosheets exhibit in-plane proton conductivities of ~1.02 and 1.18 S cm$^{-1}$ at 100% relative humidity and 90 °C, respectively, which are over five times higher than the conductivity of Nafion, the benchmark for proton conductors.

## Results

### Synthesis and characterizations of m-$H_3Sb_3P_2O_{14}$ membranes

We synthesized m-$H_3Sb_3P_2O_{14}$ nanosheets by exfoliating $H_3Sb_3P_2O_{14}$ crystals in deionized water by stirring followed by centrifugation (Fig. 1d–i and Supplementary Fig. 1). The products are negatively charged and have excellent dispersibility in deionized water due to electrostatic repulsion (Fig. 1e and Supplementary Fig. 2).

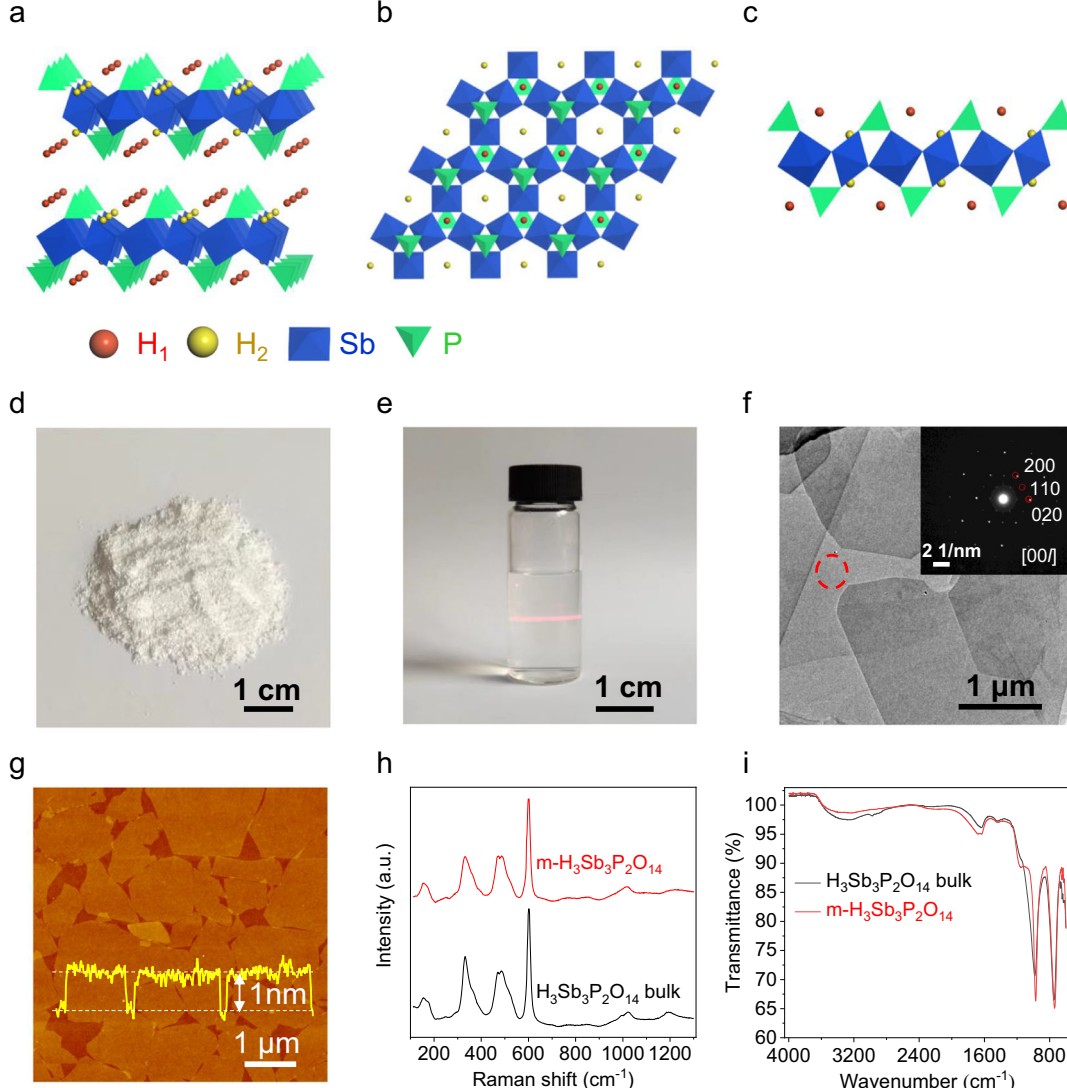

**Fig. 1 | Synthesis and characterizations of m-$H_3Sb_3P_2O_{14}$ nanosheets. a–c** The structure models of $H_3Sb_3P_2O_{14}$ bulk (**a**) and monolayer along in-plane (**b**) and cross-sectional (**c**) direction, where the oxygen atoms are omitted for clarity and the protons trapped on the bridging oxygen atoms of P-O-Sb and Sb-O-Sb are marked in red and yellow, respectively. **d** Photograph of bulk $H_3Sb_3P_2O_{14}$, which exists in the form of particles of several micrometers in size (Supplementary Fig. 1). **e** Photograph of m-$H_3Sb_3P_2O_{14}$ nanosheets aqueous dispersion with the Tyndall effect. **f** TEM image of m-$H_3Sb_3P_2O_{14}$ nanosheets. Inset is the SAED pattern taken from the area indicated by the red cycle. **g** AFM image of m-$H_3Sb_3P_2O_{14}$ nanosheets, showing a thickness of ~1 nm. **h, i** Raman spectra (**h**) and FI-IR (**i**) spectra of $H_3Sb_3P_2O_{14}$ bulk and monolayers under ~30% RH and 25 °C.

Transmission electron microscopy (TEM) image and the corresponding selective-area electron diffraction (SAED) pattern confirm that they are highly crystalline atomically thin nanosheets (Fig. 1f). Atomic force microscopy (AFM) measurements show that the nanosheets are predominantly monolayers with a thickness of ~1 nm and lateral size of ~0.5–1.5 μm (Fig. 1g and Supplementary Fig. 3). X-ray photoelectron spectroscopy (XPS) measurements indicate that the chemical compositions and valent states of Sb, P and O remain unchanged before and after exfoliation (Supplementary Fig. 4). Raman spectra and Fourier transform infrared (FT-IR) spectra suggest that m-$H_3Sb_3P_2O_{14}$ nanosheets have almost the same crystal and bonding structure with their bulk crystals (Fig. 1h, i). The slight broadening of Raman peaks and small shift of FT-IR peaks for m-$H_3Sb_3P_2O_{14}$ nanosheets might be caused by the greatly reduced interlayer coupling.

Free-standing m-$H_3Sb_3P_2O_{14}$ membranes were fabricated from their aqueous dispersion by vacuum filtration. The high optical transparency indicates that they are an electronic insulating material (Fig. 2a). The cross-sectional scanning electron microscopy (SEM) image and X-ray diffraction (XRD) pattern show that the membranes

have well-ordered layered structure (Fig. 2b and Supplementary Fig. 5a). Moreover, they are highly hydrophilic (inset of Fig. 2c), with a contact angle of ~26°, enabling good water uptake under humidity atmosphere. According to the XRD pattern, the interlayer distance ($d$) is increased by ~0.3 nm compared to that of bulk crystals (0.62 nm)[24] even at 0% RH (Supplementary Fig. 5a), indicating that the nanochannels are pre-inserted with one-layer water molecules (~0.28 nm). Based on the water adsorption-desorption isotherms (Supplementary Fig. 6), additional water uptake as high as ~2.5 and 10.2 mol mol$^{-1}$ was obtained after achieving equilibrium at 30 and 100% RH, respectively. As a result, the $d$ of m-$H_3Sb_3P_2O_{14}$ membranes reaches 1.51 nm at 100% RH (Fig. 2c and Supplementary Fig. 5b), which is increased by ~0.9 nm compared to that of bulk crystals. This yields up to a trilayer water molecule network in the nanochannels.

## Proton transport behaviors of m-$H_3Sb_3P_2O_{14}$ membranes

Chrono amperometry and linear sweep voltammetry (LSV) were used to evaluate the electronic conductivity of m-$H_3Sb_3P_2O_{14}$ membranes. They show typical characteristics of electronic insulators with no

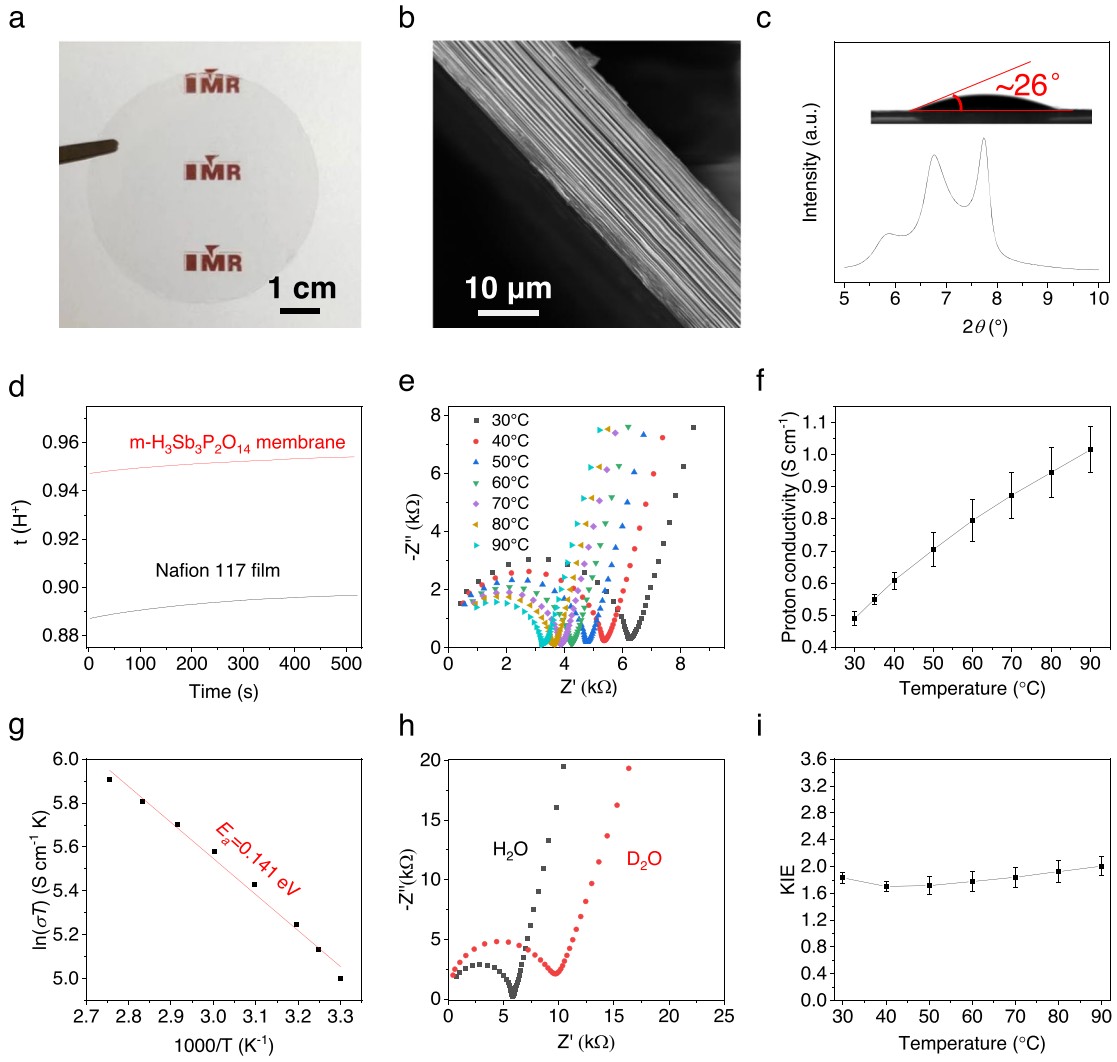

**Fig. 2 | Proton transport behaviors of m-$H_3Sb_3P_2O_{14}$ membranes.**
**a**, **b** Photograph (**a**) and cross-sectional SEM image (**b**) of a free-standing m-$H_3Sb_3P_2O_{14}$ membrane. The "IMR" LOGO in (**a**) is used with permission from the Institute of Metal Research, Chinese Academy of Sciences. **c** The XRD pattern of a m-$H_3Sb_3P_2O_{14}$ membrane at 100% RH. Inset shows the high wettability of the m-$H_3Sb_3P_2O_{14}$ membrane toward the water. **d** The proton transference number (t($H^+$)) of m-$H_3Sb_3P_2O_{14}$ membrane and Nafion 117 (DuPont company) as a function of test time. **e** Typical Nyquist plots of m-$H_3Sb_3P_2O_{14}$ membranes at different temperatures and 100% RH. **f** Temperature-dependent proton conductivities of m-$H_3Sb_3P_2O_{14}$ membranes at 100% RH. **g** Arrhenius plot of proton conductivities of m-$H_3Sb_3P_2O_{14}$ membranes at 100% RH. **h** Typical Nyquist plots for the m-$H_3Sb_3P_2O_{14}$ membranes in the presence of 100% $H_2O$ RH and 100% $D_2O$ RH at 40 °C. **i** The KIE of m-$H_3Sb_3P_2O_{14}$ membranes as a function of temperature. Error bars represent standard deviations.

detectable current at 0% RH, consistent with the high optical transparency (Fig. 2a), while a noticeable electrical current was obtained as RH increased to 100%, suggesting that the membranes are ionic conductors (Supplementary Fig. 7a–c). Such electrical behaviors are similar to those of Nafion 117 film (Supplementary Fig. 7d–f). We further measured the proton transference number (t(H$^+$)) of m-$H_3Sb_3P_2O_{14}$ membranes by H$^+$/NO$_3^-$ transference experiments. The obtained t(H$^+$) is 0.954, larger than that of Nafion 117 films (0.897) (Fig. 2d), suggesting that m-$H_3Sb_3P_2O_{14}$ membranes are pure proton conductors.

We then measured the proton conductivities of m-$H_3Sb_3P_2O_{14}$ membranes at 100% RH and different temperatures with two-electrode alternating current impedance. Before measurements, the membranes were stored at 100% RH for more than 24 hours to achieve equilibrium (Supplementary Figs. 6a, 8). In sharp contrast to the pellets made by $H_3Sb_3P_2O_{14}$ particles, m-$H_3Sb_3P_2O_{14}$ membranes show good operation stability even at 100% RH and 90 °C (Supplementary Fig. 9). They exhibit typical impedance spectra for proton conductors[27], with a semicircle and an incline spur in the high-frequency and low-frequency regions, respectively (Fig. 2e and Supplementary Fig. 10). The proton conductivities were calculated using the equation:

$$\sigma = \frac{L}{RS} \qquad (1)$$

where $\sigma$ (S cm$^{-1}$) is ion conductivity, $L$ (cm) is the length of the membrane between the blocked electrodes, $R$ ($\Omega$) is the resistance calculated from Nyquist plots, and $S$ (cm$^2$) is the cross-sectional area of the membrane. Notably, the membranes show superior in-plane proton conductivity of ~0.49 S cm$^{-1}$ even at a low temperature of 30 °C, and it increases monotonically to ~1.02 S cm$^{-1}$ as the temperature increases to 90 °C (Fig. 2f). This is different from Nafion membrane, which shows decreased proton conductivity when the temperature is higher than 80 °C[17]. These in-plane proton conductivities are over 5 times higher than the conductivity of Nafion under the same conditions[17].

We calculated the activation energy ($E_a$) for proton transport in m-$H_3Sb_3P_2O_{14}$ membranes by using the equation

$$\ln(\sigma T) = \ln \sigma_0 - \frac{E_a}{RT} \qquad (2)$$

where $\sigma_0$ is a preexponential factor, $R$ is the gas constant, and $T$ is temperature. This yields a value of ~0.141 eV (Fig. 2g), which is almost the same as the theoretical upper limit (0.09–0.13 eV)[28], indicating the very small obstacles of proton transport in m-$H_3Sb_3P_2O_{14}$ membranes. We further studied the kinetic isotope effect (KIE) of proton transport in the membranes, which yields a similar value of ~1.70 – 2.00 over the temperature 30–90 °C (Fig. 2h, i). Both $E_a$ and KIE values suggest that the proton transport in m-$H_3Sb_3P_2O_{14}$ membranes is governed by the Grotthuss mechanism ($E_a \le 0.4$ eV, KIE $\ge 1.4$)[28], in which the protons hop among the hydrogen-bonded water networks in the 2D nanochannels.

We further studied the proton transport of m-$H_3Sb_3P_2O_{14}$ membranes at low temperatures, the influence of membrane thickness and nanosheet size on the proton conductivity, and the stability of the membranes. Notably, the m-$H_3Sb_3P_2O_{14}$ membranes still show high in-plane conductivity from 0.41 S cm$^{-1}$ to 0.01 S cm$^{-1}$ at low temperature from 20 to −30 °C (Supplementary Fig. 11). In contrast, Nafion membrane shows conductivity of 0.063–0.007 S cm$^{-1}$ at the same temperature (Supplementary Fig. 11). Moreover, both the membrane thickness and nanosheet size have negligible influence on the proton conductivity (Supplementary Figs 3b, 12–14). For instance, the proton conductivity of the m-$H_3Sb_3P_2O_{14}$ membranes is ~0.52, 0.53, and 0.56 S cm$^{-1}$ for a membrane thickness of ~7, 11, and 15 μm, respectively, at 30 °C and 100% RH (Supplementary Fig 12). It also remains similar

value of ~0.49, 0.52, 0.52, and 0.53 S cm$^{-1}$ for the m-$H_3Sb_3P_2O_{14}$ membranes assembled by the nanosheets with sizes of 0.92, 0.38, 0.22, and 0.15 μm, respectively (Supplementary Figs. 3b, 13, 14). In addition, m-$H_3Sb_3P_2O_{14}$ membranes are very stable in a strong acid environment. They show almost the same Raman, FT-IR spectra, and proton conductivity before and after immersing in 10 M $H_2SO_4$ for 12 days (Supplementary Fig. 15).

## Origin of the superior proton conductivity in m-$H_3Sb_3P_2O_{14}$ membranes

We studied the influence of the thickness of $H_3Sb_3P_2O_{14}$ nanosheets on the proton transport in their assembled membranes, and found that the reduced nanosheet thickness plays a key role in boosting the proton transport (Fig. 3, Supplementary Figs. 3, 16–21, and Supplementary Discussion 1). The pellets made by $H_3Sb_3P_2O_{14}$ particles have engineering issues and may not have highly oriented nanochannels as m-$H_3Sb_3P_2O_{14}$ membranes do. To rule out the influence of channel orientation on the proton transport, we synthesized $H_3Sb_3P_2O_{14}$ membranes using nanosheets with different average thicknesses (~1.0, 1.4, 3.1, and 8.6 nm) (Supplementary Figs. 3a, 16), named as 1.0 nm- (i.e., monolayer), 1.4 nm-, 3.1 nm- and 8.6 nm-$H_3Sb_3P_2O_{14}$ membranes. The XPS, Raman, and FT-IR spectra indicate that the membranes have almost the same chemical composition, bonding, and crystal structure (Supplementary Figs. 17, 18). Moreover, they have similar average sizes from ~0.92 to ~1.27 μm (Supplementary Figs. 3b, 19). Importantly, all these membranes have no engineering issue. The cross-sectional SEM images and XRD patterns indicate that all these membranes have well-ordered layered structures (Fig. 3a–d and Supplementary Fig. 20). We quantitatively characterized the orientation degree of the nanosheets in these membranes using wide-angle X-ray scattering (WAXS), where the orientation degree is expressed by Herman's orientation factor ($f$). As shown in Fig. 3e–h, all the membranes show very similar WAXS patterns and $f$ values from 0.97 to 0.99 for the (003) peak, confirming that these membranes have highly oriented structures along (00$l$) crystal plane with almost the same orientation degree.

We then investigated the proton transport properties of the four kinds of membranes. Figure 3i show that the proton conductivity of $H_3Sb_3P_2O_{14}$ membranes increases greatly with reducing the average thickness of $H_3Sb_3P_2O_{14}$ nanosheets. The m-$H_3Sb_3P_2O_{14}$ membranes show the highest proton conductivity over the investigated temperature range at 100% RH, which is about two, three, and four times larger than that of the membranes assembled from $H_3Sb_3P_2O_{14}$ nanosheets with an average thickness of ~1.4, 3.1, and 8.6 nm, respectively. Considering almost the same structure of the four membranes, these results give strong evidence that the significantly improved proton conductivity in m-$H_3Sb_3P_2O_{14}$ membranes is mainly attributed to the reduced thickness of the nanosheets. However, all the membranes show a similar low $E_a$ of 0.13–0.15 eV (Supplementary Fig. 21a) and KIE of 1.4–2.1 over the temperature from 30 to 90 °C at 100% RH (Supplementary Fig. 21b), indicating that the proton transport in all these membranes is governed by the Grotthuss mechanism with similar hindrance. Thus, the dependence of proton conductivity on the average thickness of $H_3Sb_3P_2O_{14}$ nanosheets should be dominantly attributed to the difference in the number of active protons with high mobility in different membranes.

To further reveal the origin of the significantly improved proton transport in the m-$H_3Sb_3P_2O_{14}$ membranes, we studied the proton transport behaviors in m-$H_3Sb_3P_2O_{14}$ nanosheets, membranes, and bulk $H_3Sb_3P_2O_{14}$ at 100% RH and 30 °C by solid-state nuclear magnetic resonance (NMR) spectroscopy (Fig. 3j, k, Supplementary Figs. 22, 23, and Supplementary Discussion 1). Notably, m-$H_3Sb_3P_2O_{14}$ membranes and nanosheets show almost identical $^1H$ and $^{31}P$ NMR spectra, both of which substantially differ from those of bulk $H_3Sb_3P_2O_{14}$. These results suggest that the chemical environment and dynamics of protons change significantly as the thickness of $H_3Sb_3P_2O_{14}$ is reduced to

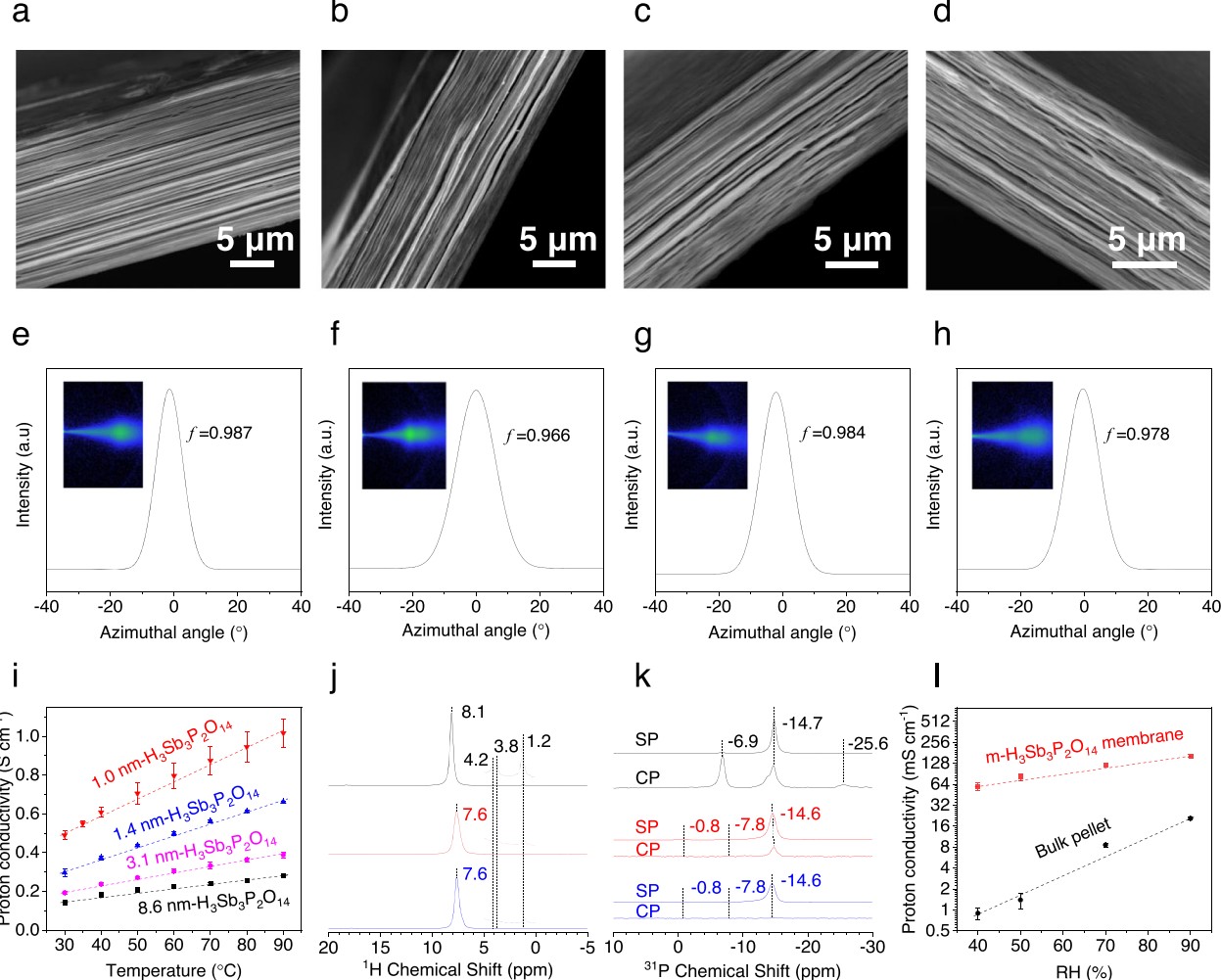

**Fig. 3 | Origin of the superior proton conductivity of m-$H_3Sb_3P_2O_{14}$ membranes. a−d** Cross-sectional SEM images of 1.0 nm-$H_3Sb_3P_2O_{14}$ (m-$H_3Sb_3P_2O_{14}$) membrane (**a**), 1.4 nm-$H_3Sb_3P_2O_{14}$ membrane (**b**), 3.1 nm-$H_3Sb_3P_2O_{14}$ membrane (**c**), and 8.6 nm-$H_3Sb_3P_2O_{14}$ membrane (**d**). **e−h** The corresponding azimuthal scan profiles for the (003) peak with the derived $f$ values. The insets are the corresponding WAXS patterns for an incident Cu-Kα X-ray beam parallel to the membrane plane. **i** Proton conductivities at different temperatures for the above four membranes. Dashed lines are guides for the eye. **j, k** $^1$H spin-echo MAS NMR spectra (**j**) and $^{31}$P MAS NMR spectra (**k**) of bulk $H_3Sb_3P_2O_{14}$ (black line), m-$H_3Sb_3P_2O_{14}$ membrane (red line) and m-$H_3Sb_3P_2O_{14}$ nanosheets (blue line) at 100% RH and 30 °C. The insets in (**j**) are the zoom-in spectra ranging from 5 to 0 ppm. **l** Comparison of the proton conductivities of m-$H_3Sb_3P_2O_{14}$ membranes and $H_3Sb_3P_2O_{14}$ bulk pellets at RH ≤90% and 30 °C. Dashed lines are guides for the eye. Error bars represent standard deviations.

monolayer, but they remain nearly unchanged when the m-$H_3Sb_3P_2O_{14}$ nanosheets are assembled into membranes with $d$ of 1.51 nm.

Figure 3j shows that for bulk $H_3Sb_3P_2O_{14}$, m-$H_3Sb_3P_2O_{14}$ membranes, and nanosheets, their $^1$H spin-echo magic angle spinning (MAS) NMR spectra are dominated by a strong peak at 8.1, 7.6, and 7.6 ppm, respectively. Such dominant and sharp $^1$H signals are attributed to those active protons with high mobility in the water between the $Sb_3P_2O^{3-}_{14}$ layer frameworks. The downshifts of $^1$H signals for m-$H_3Sb_3P_2O_{14}$ nanosheets and membranes are due to the higher water content (Supplementary Fig. 6)[29]. It is worth noting that several weak $^1$H signals at 4.2, 3.8, and 1.2 ppm are also observed for bulk $H_3Sb_3P_2O_{14}$. Suggested by the observed $^1$H-$^{31}$P dipolar coupling in $^1$H{$^{31}$P} symmetry-based resonance-echo double-resonance (S-REDOR) experiments (Supplementary Fig. 23), the protons at 4.2 and 1.2 ppm are less mobile and attached to the $Sb_3P_2O^{3-}_{14}$ layers. The $^1$H-$^{31}$P dipolar coupling exists between the fixed and spatially close $^1$H and $^{31}$P atoms. In contrast, no immobile protons attaching to the $Sb_3P_2O^{3-}_{14}$ layers are observed for the m-$H_3Sb_3P_2O_{14}$ membrane and nanosheets, as evidenced by the absence of $^1$H-$^{31}$P dipolar coupling (Supplementary Fig. 23).

The $^{31}$P NMR spectra were also acquired using both single-pulse (SP) and $^{31}$P{$^1$H} cross-polarization (CP) experiments (Fig. 3k). The $^{31}$P SP spectra show the signals of all the phosphates groups in samples, while the $^{31}$P{$^1$H} CP spectra reveal the connectivity of phosphate groups to protons. Specifically, signal in $^{31}$P{$^1$H} CP NMR spectra appears only if the phosphate group and proton possess low mobility, and are in proximity to each other. According to the $^{31}$P SP NMR spectra, the $^{31}$P signals of three samples are mainly from the $Q^3$ sites[30] (phosphate group with three bridging oxygen) forming hydrogen bonds with $H_2O$. Notably, for bulk $H_3Sb_3P_2O_{14}$, $^{31}$P{$^1$H} CP signals show both the hydrogen-bonded (from −6.9 to −14.7 ppm) and protonated (−25.6 ppm) $Q^3$ sites[30], indicating extensive bindings between protons and the $Sb_3P_2O^{3-}_{14}$ layers. In contrast, for m-$H_3Sb_3P_2O_{14}$ nanosheets, all the $^{31}$P NMR signals that appeared in the SP NMR spectrum are not observed in the $^{31}$P{$^1$H} CP NMR spectrum. Similarly, only a weak $^{31}$P{$^1$H} CP signal at −14.6 ppm is presented in m-$H_3Sb_3P_2O_{14}$ membranes. This weak signal indicates the existence of a small amount of rigid hydrogen bonds at the layer surface, which arise from the nanosheets stacked with a small $d$ in the membranes (Supplementary Fig. 5b). The differences in $^{31}$P{$^1$H} CP spectra of the three materials suggest that the

interactions between protons and $Sb_3P_2O^{3-}_{14}$ layer frameworks in m-$H_3Sb_3P_2O_{14}$ membranes and nanosheets are significantly reduced compared to bulk $H_3Sb_3P_2O_{14}$, which results in substantial increase in the number of active protons and consequently the superhigh proton conductivities of m-$H_3Sb_3P_2O_{14}$ membranes.

In addition to providing hydrogen-bonded networks for proton hopping, inserting water molecules into the nanochannels can decouple the adjacent layers by increasing $d$ and weaken the interactions between protons and $Sb_3P_2O^{3-}_{14}$ layer frameworks. Therefore, the proton transport of m-$H_3Sb_3P_2O_{14}$ membranes strongly depends on the RH. XRD results show that $d$ gradually increases from 0.94 to 1.13 nm as the RH increases from 0 to 93% RH (Supplementary Fig. 24a–c), indicating the inserting of about one-layer water molecules (bilayer in total). Importantly, $d$ rapidly increases to 1.51 nm when further increasing RH to 100%, attributed to the inserting of an additional water layer, totally up to trilayer water in the nanochannels. As a result, the proton conductivity of m-$H_3Sb_3P_2O_{14}$ membranes gradually increases with RH below 95% RH and then shows a sharp increase until 100% RH (Supplementary Fig. 25). Different from m-$H_3Sb_3P_2O_{14}$ membranes, only bilayer water molecules are inserted in bulk $H_3Sb_3P_2O_{14}$ at 100% RH (Supplementary Fig. 24d–f), which is not sufficient for decoupling the adjacent layers as confirmed by the NMR measurement results.

Surprisingly, the reduced proton/$Sb_3P_2O^{3-}_{14}$ layer interaction in m-$H_3Sb_3P_2O_{14}$ enables its membranes more pronounced advantages over bulk $H_3Sb_3P_2O_{14}$ pellets at low RHs. Notably, under 40% RH, the m-$H_3Sb_3P_2O_{14}$ membrane shows a proton conductivity ~66 times larger than that of the bulk pellets. In contrast, under 90% RH, the proton conductivity is increased approximately eight times (Fig. 3l). Moreover, under low RHs, the proton conductivities of m-$H_3Sb_3P_2O_{14}$ membranes are about one order of magnitude higher than those of Nafion[17], which is larger than the improvement under high RHs.

## Discussion

Our work illustrates a strategy for improving the concentration of proton carriers in the nanochannels to boost proton transport by using monolayer solid acids. To demonstrate the versatility of this strategy, m-$HSbP_2O_8$ (with only one kind of protons) nanosheets assembled membranes were also fabricated (Fig. 4a–g and Supplementary Figs. 26, 27), which exhibit a highly ordered lamellar structure with excellent hydrophilicity and $d$ of 1.54 nm (trilayer water intercalation) at 100% RH. Similar to m-$H_3Sb_3P_2O_{14}$ membranes, their proton conductivities are substantially improved by ~30 times compared with the bulk $HSbP_2O_8$ pellets (Fig. 4h). Significantly, they show a high in-plane proton conductivity of ~1.18 S cm$^{-1}$ with a low $E_a$ of 0.127 eV at 100% RH and 90 °C (Fig. 4i and Supplementary Fig. 28), which is better than that of m-$H_3Sb_3P_2O_{14}$ membranes. As an example, using m-$HSbP_2O_8$ membranes and MXene membranes as electrolytes and electrodes, respectively, we fabricated a high-performance all-2D flexible solid-state micro-supercapacitor with rate capability up to 300 mV s$^{-1}$ and volumetric energy density of ~18.5 mWh cm$^{-3}$ (Supplementary Methods, Supplementary Figs. 29–32, Supplementary Movie 1, and Supplementary Discussion 2), showing the great potential of m-$HSbP_2O_8$ membranes for energy applications.

Considering the availability of a vast number of solid acids, we envision that the proton transport performance could be further improved by selecting suitable layered solid acids. Such membranes not only have great potentials in the applications of supercapacitors, batteries, sensors, and neuromorphic computing, but also provide possibilities for discovering unusual nanofluidic phenomena in the nano-confined capillaries, which will expand the unique properties of 2D materials at the monolayer limit in addition to the fascinating electronic, optical, thermal and mechanical properties.

## Methods

### Materials

Nitric acid (Analytical reagent (AR), $HNO_3$) were purchased from Sinopharm Chemical Reagents (Shanghai) Co., Ltd. Ammonium dihydrogen phosphate (AR, 99%, $NH_4H_2PO_4$), antimony trioxide ($Sb_2O_3$), strontium chloride (AR, 99.5%, $SrCl_2$), magnesium chloride (AR, $MgCl_2$), lithium chloride (AR, >99%, LiCl) and phosphorus pentoxide (AR, >98%, $P_2O_5$) were purchased from Shanghai Aladdin Biochemical Technology Co., Ltd. Magnesium nitrate (AR, $Mg(NO_3)_2$) was purchased from Damao Chemical Reagent Factory. Potassium nitrate (ACS, >98%, $KNO_3$) was purchased from Alfa Aesar. Deuterium oxide (99.9 at% D, $D_2O$) was purchased from Beijing Innochem Science & Technology Co., Ltd.

### Synthesis of $H_3Sb_3P_2O_{14}$ nanosheets

We synthesized bulk $H_3Sb_3P_2O_{14}$ by the solid-state reaction and the ion exchange processes[23]. A mixture of $NH_4H_2PO_4$ (4.6 mmol), $Sb_2O_3$ (3.4 mmol), and $KNO_3$ (6.8 mmol) was well-mixed with agate mortar. It was heated in air to 300 °C for 10 h to decompose $NH_4H_2PO_4$, and then annealed at 1000 °C for 24 h to yield $K_3Sb_3P_2O_{14}$. The $K_3Sb_3P_2O_{14}$ powders (-1.60 g) were dispersed in a -200 mL $HNO_3$ solution (8 M) at 50 °C for >24 h to exchange the potassium ions by protons. In the ion exchange process, the white sediments were filtrated on polyethersulfone (pore size, 0.22 μm) membrane filters and the $HNO_3$ solution was refreshed at least three times to thoroughly exchange potassium ions by protons, producing high-purity $H_3Sb_3P_2O_{14}$ phase. The obtained $H_3Sb_3P_2O_{14}$ powders were rinsed in deionized water to remove residual $HNO_3$. After rinsing, the $H_3Sb_3P_2O_{14}$ powders were exfoliated in deionized water by stirring at 30 °C for at least 12 h. Differential centrifugation was used to obtain $H_3Sb_3P_2O_{14}$ nanosheets with different average thicknesses. The dispersion was first centrifuged at 13,000 × $g$ ($g$ = 9.80 m s$^{-2}$) for 10 min to obtain m-$H_3Sb_3P_2O_{14}$ nanosheet dispersion (supernatant). Then, the sediment was again dispersed in water, followed by centrifugation at 7312 × $g$ for 30 minutes, which yielded $H_3Sb_3P_2O_{14}$ with an average thickness of -1.4 nm in the supernatant. Similar to the above differential centrifugation steps, 3250 × $g$ and 1106 × $g$ relative centrifugal forces were used to yield aqueous dispersions of $H_3Sb_3P_2O_{14}$ nanosheets with an average thickness of -3.1 nm and 8.6 nm, respectively. The m-$H_3Sb_3P_2O_{14}$ nanosheets with average lateral sizes of -0.38, 0.22, and 0.15 μm were synthesized at ultrasonic power and time of 160 W and 5 min, 160 W and 20 min, and 320 W and 20 min, respectively.

### Synthesis of m-$HSbP_2O_8$ nanosheets

We synthesized bulk $HSbP_2O_8$ as follows[25,31]. A mixture of $NH_4H_2PO_4$ (9.24 mmol), $Sb_2O_3$ (2.310 mmol), and $KNO_3$ (4.6 mmol) was well-mixed with an agate mortar. It was first annealed at 200 °C for 4 h and then at 850 °C for 15 h. The product was ground and then acidified by 8 M $HNO_3$ at 50 °C for >24 h. In the acidification process, the $HNO_3$ solution was refreshed three times to completely exchange potassium ions with protons. The obtained bulk $HSbP_2O_8$ was rinsed in deionized water to remove residual $HNO_3$. After rinsing, the bulk $HSbP_2O_8$ was exfoliated in deionized water by stirring at 30 °C for at least 12 h. Finally, the resulting dispersion was centrifuged at 13,000×$g$ for 10 min to obtain m-$HSbP_2O_8$ nanosheet aqueous dispersion (supernatant).

### Fabrication of $H_3Sb_3P_2O_{14}$ and $HSbP_2O_8$ nanosheet membranes

The vacuum filtration method was used to fabricate $H_3Sb_3P_2O_{14}$ and $HSbP_2O_8$ nanosheet membranes. The $H_3Sb_3P_2O_{14}$ and $HSbP_2O_8$ nanosheets aqueous dispersions were filtrated under vacuum with polycarbonate (PC, Whatman™, pore size 0.2 μm) membranes as a filter. After drying in a vacuum at 50 °C for 24 h, the products were peeled off from the filters to obtain free-standing membranes.

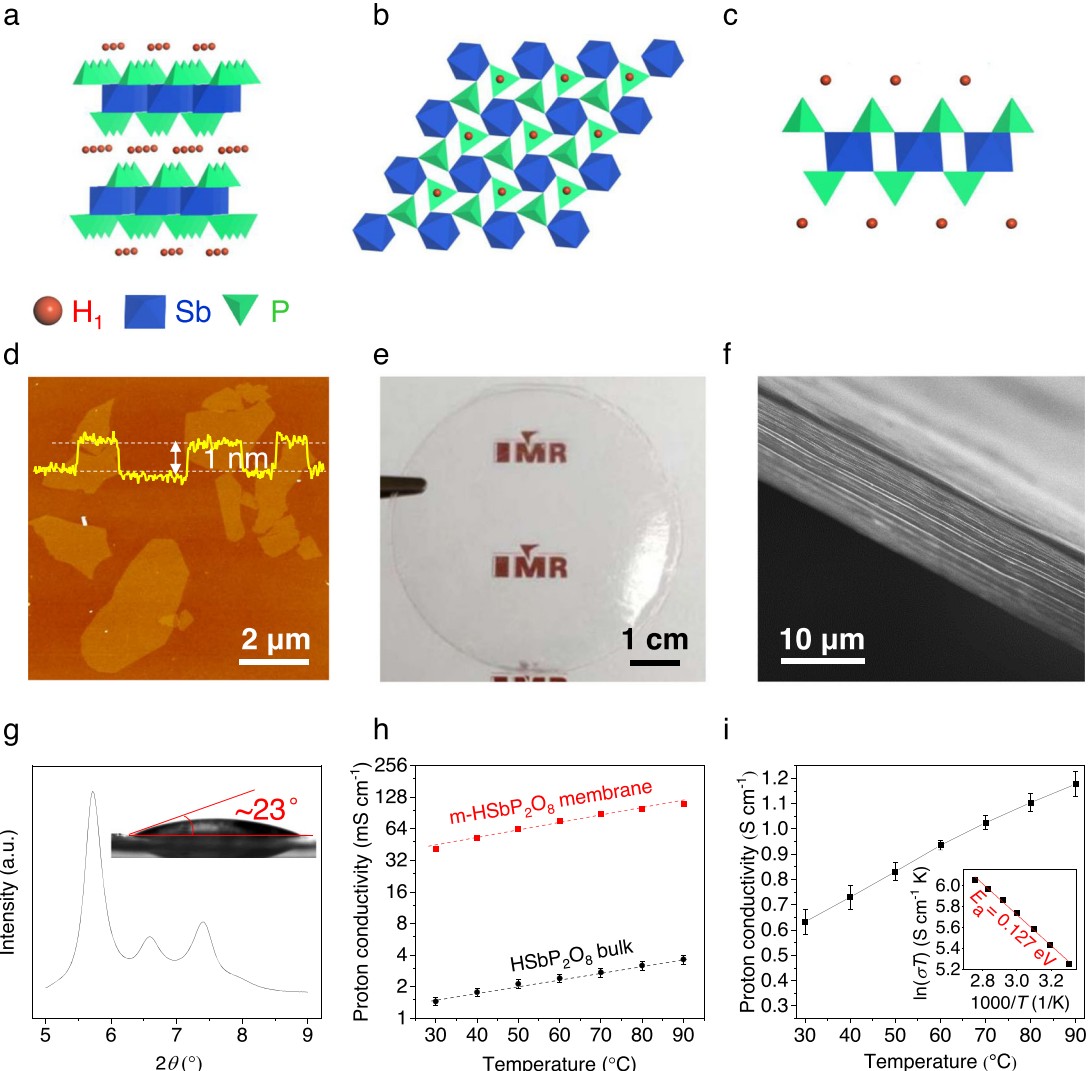

**Fig. 4 | Structure and proton transport behaviors of m-HSbP$_2$O$_8$ membranes.** **a–c** The structure models of HSbP$_2$O$_8$ bulk (**a**) and monolayer along in plane (**b**) and cross-sectional (**c**) direction, where the oxygen atoms are omitted for clarity, and the protons trapped on the bridging oxygen atoms of P-O-Sb are marked in red. **d** AFM image of m-HSbP$_2$O$_8$ nanosheets, showing a thickness of ~1 nm. **e**, **f** Photograph (**e**) and the cross-sectional SEM image (**f**) of a free-standing m-HSbP$_2$O$_8$ membrane. The "IMR" LOGO in (**e**) is used with permission from the Institute of Metal Research, Chinese Academy of Sciences. **g** The XRD pattern of m-HSbP$_2$O$_8$ membrane at 100% RH. Inset showing the wettability of m-HSbP$_2$O$_8$ membrane toward the water. **h** Comparison of proton conductivities of m-HSbP$_2$O$_8$ membranes and HSbP$_2$O$_8$ bulk at different temperatures and 70% RH. Dashed lines are guides for the eye. **i** Temperature dependence of the proton conductivities and the corresponding Arrhenius plot (inset) of m-HSbP$_2$O$_8$ membranes at 100% RH. Error bars represent standard deviations.

## Water vapor adsorption-desorption measurements

The water vapor adsorption-desorption of bulk H$_3$Sb$_3$P$_2$O$_{14}$ and m-H$_3$Sb$_3$P$_2$O$_{14}$ membranes was measured in a Dynamic Vapor Sorption (DVS Intrinsic, Surface Measurement Systems Ltd., UK) equipped with an ultrasensitive microbalance of mass resolution (±1 μg) and a high precision RH control system (±1%) at 303 K. Prior to measurements, both the membranes and bulk samples were vacuumed at 393 K for more than 48 h. The equilibration time for adsorption-desorption at each RH was determined by the value of $d$m/$d$t (<0.0002% min$^{-1}$).

## Electronic conductivity measurements

Chrono amperometry and LSV measurements were performed to evaluate the electronic conductivities of m-H$_3$Sb$_3$P$_2$O$_{14}$ membranes and Nafion 117 film (DuPont Company) on an electrochemical workstation (Autolab M204 (Metrohm), potentiostat–galvanostat). To void the ionic current, before testing, both samples were dried at 80 °C in vacuum with P$_2$O$_5$ as a desiccant for 5 days.

## Proton transference number measurements

The proton transference numbers of m-H$_3$Sb$_3$P$_2$O$_{14}$ membranes and Nafion 117 films were measured using a two-cell apparatus[32]. Both samples were first soaked in 2.0 M HNO$_3$ solution for 72 h at 25 °C, and then mounted between the cells. The two chambers of the cells were filled with HNO$_3$ solutions with different concentrations (1.0–3.0 M). The open circuit potential ($E_{ocp}$) equals the value of the membrane/film potential due to the Donnan exclusion, and it was measured by connecting both cells and Ag/AgCl electrode through saturated chloride potassium bridges. The proton transference number was calculated by the following equation:

$$E_{ocp} = -(t_+ - t_-)\frac{RT}{F}\ln\frac{a_2}{a_1} \quad (3)$$

where $t_+$, $t_-$, $R$, $T$, $F$, and $\frac{a_2}{a_1}$ are proton transference number, $NO_3^-$ transference number, gas constant, temperature, faraday constant, and activity gradient, respectively.

## Proton conductivity measurements

Electrochemical impedance spectroscopy measurements were performed by using an Autolab M204 (Metrohm, PGSTAT204) electrochemical workstation with a tunable frequency range from 1 MHz to 10 Hz at a potential of 0 with 150 mV amplitude. The proton conductivities of $H_3Sb_3P_2O_{14}$ and $HSbP_2O_8$ based materials at different RHs and temperatures were tested in a climatic test chamber (LabEvent, Weiss-Voetsch Environmental Testing Instruments (Taicang) Co., Ltd.). The 100% RH atmosphere was created by water sealed in a dish. For testing, the free-standing $H_3Sb_3P_2O_{14}$ and $HSbP_2O_8$ nanosheet membranes were cut into strips ($40.0 \times 5.0$ mm$^2$) and put on the quartz holder with a square hole ($20.0 \times 20.0$ mm$^2$). For bulk $H_3Sb_3P_2O_{14}$ and $HSbP_2O_8$, they were ground to powders by agate mortar and then pressed under a pressure of ~0.76 GPa to obtain pellets, and finally, the pellets were cut into strips ($25.0 \times 5.0$ mm$^2$). The platinum strips (purity: 99.99%, ZhongNuo Advanced Material (Beijing) Technology Co., Ltd.) were attached to both ends of the membranes/bulk pellets and used as blocked electrodes. The thicknesses of the tested membranes and pellets at certain RHs were measured by micrometer (MDC-25PX, accuracy ±1 μm, Mitutoyo Corporation (Japan)). At least three different samples were measured for each kind of membranes/pellets.

## KIE measurements

Before measurements, the m-$H_3Sb_3P_2O_{14}$ membranes were dried at 80 °C in a vacuum with $P_2O_5$ as a desiccant for 5 days to exclude the influence of water vapor. Then, the membranes were stored in the vapor of $D_2O$ for more than 24 h. EIS measurements were performed at different temperatures ranging from 30 to 90 °C.

## NMR measurements

The $^1H$ spin-echo and $^{31}P$ SP/CP NMR spectra were acquired in a 14.1 T Bruker AVIII 600 NMR spectrometer, using a 4 mm triple-resonance MAS probe at 12 kHz spinning frequency. Excitation of the $^1H$ and $^{31}P$ signals used a 90° pulse with a pulse length of 9.1 and 6.2 μs, respectively. For $^1H$ spin-echo NMR experiments, the echo duration was 83.3 μs. For $^{31}P\{^1H\}$ CP NMR experiments, the CP contact used 24.8 and 28.6 kHz radiofrequency (RF) amplitude for $^1H$ and $^{31}P$ nuclei, respectively. For $^1H\{^{31}P\}$ S-REDOR experiments, the RF amplitude of $SR4_1^2$ dipolar recoupling on the $^1H$ channel was 24 kHz (twice the spinning frequency), and the recouping time was 5.0 ms. The reference spectra ($S_0$) were obtained by omitting the refocusing pulse at the $^{31}P$ channel, while the spectra containing $^1H$-$^{31}P$ dipolar dephase (decreasing signal intensity) were obtained by applying the $^{31}P$ refocusing pulse. In the $^1H$-$^1H$ 2D exchange experiments of the $H_3Sb_3P_2O_{14}$ bulk sample, the excitation 90° pulse was 9.8 μs, with 10.0 ms mixing time. The 2D $^1H$-$^1H$ exchange NMR spectra (EXSY) were acquired in a 9.4 T Bruker AVANCE III HD 400 NMR spectrometer, operating at a 4 mm HX MAS probe at 12 kHz spinning frequency. The $^1H$ pulse field gradient (PFG) NMR spectra were acquired in a wide-bore Bruker AVIII 600 spectrometer with a 14.1 T magnet, using a 5 mm PFG probe. In $^1H$ PFG experiments, the RF amplitude for $^1H$ excitation was 18 kHz. In each experiment, the diffusion time was 8.0 ms, the duration of gradient pulses was 1.0 ms, with the gradient strength increase from 36 to 999.61 in 16 steps.

## XRD measurements at different RHs

To measure the XRD patterns of m-$H_3Sb_3P_2O_{14}$ membranes and bulk $H_3Sb_3P_2O_{14}$ pellets at different RHs, they were first dried at 80 °C in vacuum ($P_2O_5$ as a desiccant) for 5 days and then stored in a sealed dish with different saturated salt slurries including LiCl (11%), MgCl$_2$ (33%), Mg(NO$_3$)$_2$ (53%), SrCl$_2$ (70%), KNO$_3$ (93%) at 25 °C for at least 72 h. To achieve 0% RH, we put excess $P_2O_5$ as a desiccant inside the sealed dish. The 100% RH atmosphere was created by water sealed in the dish.

## Structure characterizations

The morphology and structure of $H_3Sb_3P_2O_{14}$ and $HSbP_2O_8$ nanosheets, membranes and their counterpart bulks were characterized by XRD (Rigaku diffractometer with Cu-Kα radiation between 5°–45° and an incident wavelength of 0.154056 nm), SEM (Verios G4 UC), TEM (FEI Tecnai T20, 120 kV), Helium ion microscope (HIM, Zeiss Orion Nano-Fab), and AFM (Bruker Multimode 8). XPS measurements were performed in an ESCALAB 250 spectrometer using monochromatic Al Kα radiation (1486.6 eV). The hydrophilicity of the membranes was analyzed at 298 K using a contact angle analyzer (Dataphysics, OCA20). The Raman spectra were recorded with JY Labram HR 800 using a 532 nm laser. The FT-IR spectra were acquired on a Smart OMNI-Transmission spectrometer (Thermo Scientific). Zeta-potential was measured with Malvern Zetasizer Nano-ZS90. WAXS measurements were performed on Xeuss 3.0 HR (Xenocs) SAXS/WAXS system using an incident Cu-Kα X-ray beam parallel to the membrane plane. The distance between the sample and the detector was 20.0 cm. For measurements, the membranes were cut into 5.0-mm-wide and 15.0-mm-long strips. The scattering patterns were collected by a Pilatus 3 R 300 K detector. The orientation degree of $H_3Sb_3P_2O_{14}$ nanosheets in the membranes was quantified by using Herman's orientations factor ($f$), which was defined as follows[33,34],

$$f = \frac{3\langle\cos^2\phi\rangle - 1}{2} \tag{4}$$

where $\langle\cos^2\phi\rangle$ is the average value of the square of the cosine of the azimuthal angle for the (003) peak of $H_3Sb_3P_2O_{14}$ membranes, which was calculated as follows,

$$\langle\cos^2\phi\rangle = \frac{\int_0^{\pi/2} I(\phi)\sin\phi\cos^2\phi\, d\phi}{\int_0^{\pi/2} I(\phi)\sin\phi\, d\phi} \tag{5}$$

where $I(\phi)$ is the peak intensity at an azimuthal angle of $\phi$.

## Data availability

The authors declare that the experimental data supporting the results of this study can be found in the paper and its Supplementary Information file. The detailed data for the study is available from the corresponding author upon request.

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

## Acknowledgements

We thank Z. B. Liu and Q. Wang for the help on TEM measurements and X. Yu for the discussions on MSCs fabrication. This work was supported by the National Natural Science Foundation of China (Nos. 52188101 and 21773230), the Key Research Program of Frontier Sciences of the Chinese Academy of Sciences (No. ZDBS-LY-JSC027), and the LiaoNing Revitalization Talents Program (No. XLYC2201003).

## Author contributions

W.R. conceived and supervised the project. Z.Z. carried out the experiments and measurements. L.L. carried out the NMR measurements and analyzed the NMR data under the supervision of G.H. J.F. synthesized MXene, fabricated and measured MSCs. W.R. and Z.Z. designed the experiments, analyzed the data, and wrote the manuscript with input from other authors.

## Competing interests

The authors declare no competing interests.
