## [Peer Review File · Nature Communications]

Significant enhancement of proton conductivity in solid acid at the monolayer limitREVIEWER COMMENTS

Reviewer #1 (Remarks to the Author):

The authors report ultrahigh proton conductivity in monolayer solid acid assembled membranes. With monolayer H₃Sb₃P₂O₁₄ and HSbP₂O₈ membranes as examples, they found extreme enhancement of proton conductivity at monolayer limit due to the significant reduction of interactions between protons and the layer framework. The proton conductivity of monolayer solid acid membranes is very impressive, exceeding those of the benchmark Nafion and other materials in a broad range of temperature and humidity. This work demonstrates a new concept for developing high-performance proton conductors, which have potential applications in many fields such as fuel cells and batteries. Furthermore, it provides new insight into the unique properties of 2D materials at the monolayer limit in addition to the known electronic, optical, thermal and mechanical properties. Thus, I am happy to recommend the publication of this manuscript in Nature Communications after a minor revision.

1. The authors show that the centrifugal force affects the thickness of H₃Sb₃P₂O₁₄ nanosheets. Does it affect the size? Can the size be tuned? Does the size of nanosheets affect the proton conductivity of the m-H₃Sb₃P₂O₁₄ membranes?
2. The EIS test temperature for the m-H₃Sb₃P₂O₁₄ membranes ranges from 30 to 90 C. What will happen for the proton conductivity at lower temperature? Is it still superior to that of Nafion at lower temperature?
3. Both m-H₃Sb₃P₂O₁₄ and m-HSbP₂O₈ membranes have anisotropic structure. Are the proton conductivities affected by their thickness?
4. The chemical environment in the proton exchange membrane fuel cells is very harsh (such as strong acidity). I suggest the authors check the stability of m-H₃Sb₃P₂O₁₄ membranes in strong acid.
5. The unit for the y-axis is missing in the inset of Fig. 4g.

Reviewer #2 (Remarks to the Author):

This paper reports solid acids that have high proton conductivity. The proton conductivity of H₃Sb₃P₂O₁₄ is reported > 1 S cm⁻¹ at 90 C and the authors claim that this may be a new strategy for fuel cells and other devices. The high proton conductivity of solid acids and metal phosphates is well-known from numerous previous studies. (e.g., Sossina Haile et al. Nature, 410, 910, 2001). However, despite the high conductivity of inorganic conductors, realizing the high performance of fuel cells using such materials seems to be much more challenging. After more than 20 years of research, the fuel cell performance using inorganic materials is still much inferior to the Nafion-based system. This is because there are more requirements than just conductivity for fuel cell membranes. The requirements include thin-film forming capability, stability in the presence of water, hydrogen impermeability, etc.

In this manuscript, the authors presented high conductivity, good stability under hydrated conditions, and film-forming properties. But they do not provide any single-cell performance. If they do not provide single-cell performance, the impact of this paper is greatly reduced. It is unfair to ask for very high fuel cell performance using these materials since the author's expertise is not the device testing. However, the authors should provide high-frequency resistance data with reasonable performance data (500-800 mW/cm²) to convince the readers that the proposed materials are promising. If the authors cannot provide such data, this material can be reported in a more material-specific journal.

Reviewer #3 (Remarks to the Author):

The work has not been logically carried out, presented and compared with other materials. It will be necessary to compare the materials first before any clear conclusion can be made, because the

membrane ones are more complicate which involves some engineering issues. An excellent work will need to examine the single crystal's directional dependence of the proton conductivity to figure out the mechanism. The work is quite far away from the quality of NC publication.

RESPONSE TO REVIEWERS' COMMENTS

Reviewer #1:

The authors report ultrahigh proton conductivity in monolayer solid acid assembled membranes. With monolayer $\text{H}_3\text{Sb}_3\text{P}_2\text{O}_{14}$ and HSbP_2O_8 membranes as examples, they found extreme enhancement of proton conductivity at monolayer limit due to the significant reduction of interactions between protons and the layer framework. The proton conductivity of monolayer solid acid membranes is very impressive, exceeding those of the benchmark Nafion and other materials in a broad range of temperature and humidity. This work demonstrates a new concept for developing high-performance proton conductors, which have potential applications in many fields such as fuel cells and batteries. Furthermore, it provides new insight into the unique properties of 2D materials at the monolayer limit in addition to the known electronic, optical, thermal and mechanical properties. Thus, I am happy to recommend the publication of this manuscript in Nature Communications after a minor revision.

1. The authors show that the centrifugal force affects the thickness of $\text{H}_3\text{Sb}_3\text{P}_2\text{O}_{14}$ nanosheets. Does it affect the size? Can the size be tuned? Does the size of nanosheets affect the proton conductivity of the *m*- $\text{H}_3\text{Sb}_3\text{P}_2\text{O}_{14}$ membranes?

Response: We thank the reviewer very much for the valuable suggestions.

The average lateral sizes of $\text{H}_3\text{Sb}_3\text{P}_2\text{O}_{14}$ nanosheets are 0.92 μm , 0.93 μm , 1.20 μm , and 1.27 μm for a relative centrifugal force of 13000 g ($g = 9.80 \text{ m s}^{-2}$), 7312 – 13000 g, 3250 – 7312 g, and 1106 – 3250 g, respectively (Figure R1). This indicates a very little influence of centrifugal force on the lateral size of $\text{H}_3\text{Sb}_3\text{P}_2\text{O}_{14}$ nanosheets. To reveal the relationship between the proton conductivity of $\text{H}_3\text{Sb}_3\text{P}_2\text{O}_{14}$ membranes and the lateral size of $\text{H}_3\text{Sb}_3\text{P}_2\text{O}_{14}$ nanosheets, *m*- $\text{H}_3\text{Sb}_3\text{P}_2\text{O}_{14}$ nanosheets with the same thickness ($\sim 1.0 \text{ nm}$, monolayer) and different average lateral size were synthesized by controlling the ultrasound process. As shown in Figure R1a and Figure R2, the average lateral sizes of *m*- $\text{H}_3\text{Sb}_3\text{P}_2\text{O}_{14}$ nanosheets are decreased from 0.92 μm , 0.38 μm , 0.22 μm to 0.15 μm by increasing the ultrasonic power and time. We characterized the proton transport properties of the *m*- $\text{H}_3\text{Sb}_3\text{P}_2\text{O}_{14}$ membranes assembled by these four kinds of nanosheets. The proton conductivities are $0.49 \pm 0.021 \text{ S cm}^{-1}$, $0.52 \pm 0.035 \text{ S cm}^{-1}$,

$0.52 \pm 0.025 \text{ S cm}^{-1}$ and $0.53 \pm 0.020 \text{ S cm}^{-1}$ at $30 \text{ }^\circ\text{C}$ and 100% relative humidity (RH) for the $m\text{-H}_3\text{Sb}_3\text{P}_2\text{O}_{14}$ membranes assembled by the nanosheets with lateral size of $0.92 \text{ }\mu\text{m}$, $0.38 \text{ }\mu\text{m}$, $0.22 \text{ }\mu\text{m}$ and $0.15 \text{ }\mu\text{m}$, respectively (Figure R3). Moreover, the values are also almost the same at $60 \text{ }^\circ\text{C}$ and $90 \text{ }^\circ\text{C}$. These results suggest that the influence of lateral size of $m\text{-H}_3\text{Sb}_3\text{P}_2\text{O}_{14}$ nanosheets on the proton conductivity is negligible for $m\text{-H}_3\text{Sb}_3\text{P}_2\text{O}_{14}$ membranes.

We have added these data and related discussions in the revised manuscript.

Figure R1 The lateral size distributions of the $\text{H}_3\text{Sb}_3\text{P}_2\text{O}_{14}$ nanosheets obtained at a relative centrifugal force of 13000 g ($g = 9.80 \text{ m s}^{-2}$) (a), $7312 - 13000 \text{ g}$ (b), $3250 - 7312 \text{ g}$ (c), and $1106 - 3250 \text{ g}$ (d). The average lateral sizes are shown in the upper corner of each figure.

Figure R2 The lateral size distributions of the $m\text{-H}_3\text{Sb}_3\text{P}_2\text{O}_{14}$ nanosheets obtained at ultrasonic power and time of 160 W and 5 mins (a), 160 W and 20 mins (b), and 320 W and 20 mins (c). The average lateral sizes are shown in the upper corner of each figure.

Figure R3 The relationship between the proton conductivities of $m\text{-H}_3\text{Sb}_3\text{P}_2\text{O}_{14}$ membranes and the lateral sizes of $m\text{-H}_3\text{Sb}_3\text{P}_2\text{O}_{14}$ nanosheets at 100% RH and different temperatures. Error bars represent standard deviations.

2. The EIS test temperature for the $m\text{-H}_3\text{Sb}_3\text{P}_2\text{O}_{14}$ membranes ranges from 30 to 90 °C. What will happen for the proton conductivity at lower temperature? Is it still superior to that of Nafion at lower temperature?

Response: We thank the reviewer very much for the valuable suggestions.

The proton conductivities of $m\text{-H}_3\text{Sb}_3\text{P}_2\text{O}_{14}$ membranes and Nafion117 membranes (DuPont company) were measured by EIS at low temperature from 20 to -30 °C. As shown in Figure R4, the obtained proton conductivity of $m\text{-H}_3\text{Sb}_3\text{P}_2\text{O}_{14}$ membranes

ranges from $0.41 \pm 0.020 \text{ S cm}^{-1}$ to $0.01 \pm 0.001 \text{ S cm}^{-1}$, which are much higher than those of Nafion 117 membranes at the same temperature (from $0.063 \pm 0.0038 \text{ S cm}^{-1}$ to $0.007 \pm 0.0010 \text{ S cm}^{-1}$). This result indicates that $m\text{-H}_3\text{Sb}_3\text{P}_2\text{O}_{14}$ membranes is superior to Nafion in proton conductivity at lower temperature.

We have added these data and related discussions in the revised manuscript.

Figure R4 The proton conductivities of $m\text{-H}_3\text{Sb}_3\text{P}_2\text{O}_{14}$ membranes and Nafion117 membranes at temperature from 20 to $-10 \text{ }^\circ\text{C}$ (a) and from -15 to $-30 \text{ }^\circ\text{C}$ (b). Error bars represent standard deviations.

3. Both $m\text{-H}_3\text{Sb}_3\text{P}_2\text{O}_{14}$ and $m\text{-HSbP}_2\text{O}_8$ membranes have anisotropic structure. Are the proton conductivities affected by their thickness?

Response: We thank the reviewer very much for the insightful comment.

We have tested the proton conductivities of $m\text{-H}_3\text{Sb}_3\text{P}_2\text{O}_{14}$ and $m\text{-HSbP}_2\text{O}_8$ membranes with different thicknesses. As shown in Figure R5a, the proton conductivity of $m\text{-H}_3\text{Sb}_3\text{P}_2\text{O}_{14}$ membranes is $0.52 \pm 0.048 \text{ S cm}^{-1}$, $0.53 \pm 0.024 \text{ S cm}^{-1}$ and $0.56 \pm 0.04 \text{ S cm}^{-1}$ for a membrane thickness of $7 \text{ }\mu\text{m}$, $11 \text{ }\mu\text{m}$ and $15 \text{ }\mu\text{m}$, respectively, at $30 \text{ }^\circ\text{C}$ and 100% RH. Moreover, such membranes also show a very little change in proton conductivity at $60 \text{ }^\circ\text{C}$ and $90 \text{ }^\circ\text{C}$. These results suggest that the thickness of $m\text{-H}_3\text{Sb}_3\text{P}_2\text{O}_{14}$ membranes has very little influence on their proton conductivity. The $m\text{-HSbP}_2\text{O}_8$ membranes show similar trend (Figure R5b). For instance, the proton conductivity is $0.68 \pm 0.005 \text{ S cm}^{-1}$, $0.67 \pm 0.010 \text{ S cm}^{-1}$ and $0.60 \pm 0.016 \text{ S cm}^{-1}$ for a membrane thickness of $5 \text{ }\mu\text{m}$, $8 \text{ }\mu\text{m}$ and $11 \text{ }\mu\text{m}$, respectively.

We have added these data and related discussions in the revised manuscript.

Figure R5 The relationships between the proton conductivities and the thicknesses of *m*-H₃Sb₃P₂O₁₄ (a) and *m*-HSbP₂O₈ (b) membranes at 100% RH and different temperatures. Error bars represent standard deviations.

4. The chemical environment in the proton exchange membrane fuel cells is very harsh (such as strong acidity). I suggest the authors check the stability of *m*-H₃Sb₃P₂O₁₄ membranes in strong acid.

Response: We thank the reviewer very much for the valuable suggestion.

The stability of *m*-H₃Sb₃P₂O₁₄ membranes in strong acid were characterized by EIS, Raman and FT-IR spectroscopy. As shown in Figure R6a, *m*-H₃Sb₃P₂O₁₄ membranes shows no obvious change in proton conductivity after immersing in 10 M H₂SO₄ for 12 days. Moreover, both the Raman and FT-IR spectra remain almost the same before and after H₂SO₄ treatment (Figure R6b,c). These results demonstrate that *m*-H₃Sb₃P₂O₁₄ membranes are very stable in strong acid environment.

We have added these data and related discussions in the revised manuscript.

Figure R6 The proton conductivities (a), Raman spectra (b), and FT-IR spectra (c) of *m*-H₃Sb₃P₂O₁₄ membranes before and after immersing in 10 M H₂SO₄ for 12 days. Error bars represent standard deviations.

5. The unit for the y-axis is missing in the inset of Fig. 4g.

Response: We have added the unit for the y-axis in the inset of Fig. 4g in the revised manuscript.

Reviewer #2:

This paper reports solid acids that have high proton conductivity. The proton conductivity of $\text{H}_3\text{Sb}_3\text{P}_2\text{O}_{14}$ is reported $> 1 \text{ S cm}^{-1}$ at 90 C and the authors claim that this may be a new strategy for fuel cells and other devices. The high proton conductivity of solid acids and metal phosphates is well-known from numerous previous studies. (e.g., Sossina Haile et al. Nature, 410, 910, 2001). However, despite the high conductivity of inorganic conductors, realizing the high performance of fuel cells using such materials seems to be much more challenging. After more than 20 years of research, the fuel cell performance using inorganic materials is still much inferior to the Nafion-based system. This is because there are more requirements than just conductivity for fuel cell membranes. The requirements include thin-film forming capability, stability in the presence of water, hydrogen impermeability, etc.

In this manuscript, the authors presented high conductivity, good stability under hydrated conditions, and film-forming properties. But they do not provide any single-cell performance. If they do not provide single-cell performance, the impact of this paper is greatly reduced. It is unfair to ask for very high fuel cell performance using these materials since the author's expertise is not the device testing. However, the authors should provide high-frequency resistance data with reasonable performance data (500-800 mW/cm²) to convince the readers that the proposed materials are promising. If the authors cannot provide such data, this material can be reported in a more material-specific journal.

Response: We thank the reviewer very much for the insightful comments and valuable suggestions.

Due to the highly anisotropic structure, *m*- $\text{H}_3\text{Sb}_3\text{P}_2\text{O}_{14}$ membranes exhibit much higher proton conductivity along in-plane direction than through-plane direction. For instance, as shown in the manuscript, the in-plane proton conductivity of the membrane is $\sim 0.38 \text{ S cm}^{-1}$ at 95% RH and 60 °C. However, the through-plane conductivity is $\sim 7.7 \times 10^{-4} \text{ S cm}^{-1}$ at the same test conditions (Figure R7). In the typical structure of a fuel cell such as PEMFC, large-area ($\sim \text{cm}^2$) proton exchange membrane is usually used to

achieve a large electricity (e.g., Jiao, K., et al, *Nature* 595, 361, 2021; Kraysberg, A., et al, *Energy Fuels* 28, 7303, 2014; Steele, B. C. H., et al, *Nature* 414, 345, 2001). As for the *m*-H₃Sb₃P₂O₁₄ membranes, the area of the membrane for proton transfer is only $\sim 10^{-3}$ cm² when we try to use its superhigh in-plane proton conductivity since the thickness of the membranes synthesized by the vacuum method is on the order of 10⁻⁴ cm, which is difficult to meet the requirement of a typical fuel cell. Moreover, the low through-plane proton conductivity limits the direct use of such membranes in fuel cell as Nafion does. The main finding of our work is that the interactions between protons and the layer frameworks in layered solid acid H_nM_nZ₂O_{3n+5} are substantially reduced at the monolayer limit, which significantly increases the concentration of active protons and consequently improves the proton conductivity of membranes dramatically. Thus, one possible way to use this material for fuel cell is to make composites using monolayer H₃Sb₃P₂O₁₄ nanosheets and Nafion, which is beyond the scope of this work and deserves thorough studies in the future.

Alternatively, to improve the impact of this work, we have used *m*-HSbP₂O₈ membrane to construct an all-2D solid-state micro-supercapacitor (MSCs) to demonstrate its practical use, which fully utilized the highly anisotropic structure and proton transport characteristic of the membranes. The superhigh proton conductivity and electronic insulating nature enable monolayer solid acid membrane an excellent H⁺ solid-state electrolyte and electrode separator in high-safety energy storage devices. The linear scan voltammetry (LSV) curves show that *m*-HSbP₂O₈ membrane features a significantly broader electrochemical stability window (~ 2.50 V vs Pt²⁺/Pt) than that of H₂SO₄ aqueous solution (~ 0.38 V vs Pt²⁺/Pt at 3.0 M) (Figure R8), the most commonly used H⁺ electrolyte. We fabricated flexible all-2D H⁺ solid-state MSCs by using *m*-HSbP₂O₈ membranes and MXene membranes as solid-state electrolyte (0.49 S cm⁻¹ at room temperature) and electrodes, respectively (Figure R9a,b). MXene is an excellent electrode material with high capacitance and easy to assembled into well-ordered layered membranes (e.g., Ghidui, M., et al, *Nature* 516, 78, 2014; Anasori, B., et al, *Nat. Rev. Mater.* 2, 16098, 2017). In such devices, *m*-HSbP₂O₈ and MXene membranes construct interconnected nanochannels with the same orientation (Figure R9b,c),

ensuring small obstacles of H^+ transport in the whole devices. For comparison, we also fabricated MSCs with the same electrodes using 3M H_2SO_4 solution, 3M H_2SO_4 /Poly(vinyl alcohol) (PVA) gel, solid Nafion and graphene oxide (GO) membranes as electrolytes, respectively, which have ionic conductivities of ~ 0.71 , 0.49, 0.04 and 0.01 $S\ cm^{-1}$ at room temperature.

Figure R9d shows that the *m*-HSbP₂O₈-MXene MSCs exhibit good capacitive behaviors, where the cyclic voltammetry (CV) curves maintain a rectangular-like shape with a capacitance over 1.9 $mF\ cm^{-2}$ at a superhigh scan rate of 300 $mV\ s^{-1}$, which is similar to the devices using H_2SO_4 and H_2SO_4 /PVA as electrolytes (Figure R10a,b). In contrast, the Nafion-MXene and GO-MXene MSCs show poor electrochemical performances (Figure R10c,d). Despite the similar interconnected proton transport nanochannels in GO-MXene MSCs, they show a much lower capacitance of 0.37 $mF\ cm^{-2}$ at 300 $mV\ s^{-1}$. Importantly, the *m*-HSbP₂O₈-MXene MSCs output a high operating voltage (0.9 V) (Figure R9e), which is over 1.5 times larger than those of the MSCs using other electrolytes, ranging from 0.4 V for GO membrane to 0.6 V for H_2SO_4 /PVA gels (Figure R9f, Figure R11). As a result, the *m*-HSbP₂O₈-MXene MSCs show similarly high volumetric energy densities of 18.5 – 18.3 $mWh\ cm^{-3}$ with the corresponding power densities in the range of 1.7 – 6.8 $W\ cm^{-3}$ (Figure R9g). These performances are significantly better than those of H_2SO_4 -, H_2SO_4 /PVA-, Nafion-, and GO-MXene MSCs with the same electrodes. Furthermore, such all-2D MSCs can also be used to power the electronic devices even under repeated bending without packaging (Figure R9h,i, Supplementary movie 1), demonstrating the great potential of *m*-HSbP₂O₈ membranes for practical use in flexible solid-state micro energy storage devices.

We have added these data and related discussions in the revised manuscript.

Figure R7 The through-plane conductivities of $m\text{-H}_3\text{Sb}_3\text{P}_2\text{O}_{14}$ membranes at 95% RH and different temperatures from 30 °C to 60 °C.

Figure R8. The LSV curves of fully hydrated $m\text{-HSbP}_2\text{O}_8$ membrane and 3.0 M H_2SO_4 solution at the polarization scanning of 2 mV s^{-1} .

Figure R9 Demonstration of all-2D flexible solid-state $m\text{-HSbP}_2\text{O}_8\text{-MXene}$ MSCs. **a**, Photograph of a $m\text{-HSbP}_2\text{O}_8\text{-MXene}$ MSC, showing good flexibility. **b**, Cross-sectional SEM images and the corresponding EDS mappings of the $m\text{-HSbP}_2\text{O}_8$ membrane (top) and MXene membrane (bottom). Scale bar, 5 μm . **c**, Illustration of the interconnected nanochannels in $m\text{-HSbP}_2\text{O}_8\text{-MXene}$ MSCs. The red balls surrounded by blue rings represent hydrated protons. **d**, Typical CV curves of $m\text{-HSbP}_2\text{O}_8\text{-MXene}$ MSCs at the scan rate from 30 mV s^{-1} to 500 mV s^{-1} . **e**, The galvanostatic charge-discharge (GCD) curves of the $m\text{-HSbP}_2\text{O}_8\text{-MXene}$ MSCs at different current densities. **f**, Comparison of the operating voltages of $m\text{-HSbP}_2\text{O}_8\text{-}$, $\text{H}_2\text{SO}_4\text{-}$, $\text{H}_2\text{SO}_4/\text{PVA-}$, Nafion-, and GO-MXene MSCs. **g**, The Ragone plots showing the volumetric energy and power densities of $m\text{-HSbP}_2\text{O}_8\text{-}$, $\text{H}_2\text{SO}_4\text{-}$, $\text{H}_2\text{SO}_4/\text{PVA-}$, Nafion-, and GO-MXene MSCs. **h,i**, Demonstration of the applications of $m\text{-HSbP}_2\text{O}_8\text{-MXene}$ MSCs, where five packaging-free devices were connected in series to light up a “IMR” LOGO assembled by 56 LEDs (**i**).

Figure R10. The CV curves of H₂SO₄-MXene (a), H₂SO₄/PVA-MXene (b), Nafion-MXene (c), and GO-MXene (d) MSCs at the scan rate from 30 to 500 mV s⁻¹.

Figure R11. The GCD curves of the H₂SO₄-MXene (a), H₂SO₄/PVA-MXene (b), Nafion-MXene (c), and GO-MXene (d) MSCs at different current densities.

Reviewer #3:

The work has not been logically carried out, presented and compared with other materials. It will be necessary to compare the materials first before any clear conclusion can be made, because the membrane ones are more complicate which involves some engineering issues. An excellent work will need to examine the single crystal's directional dependence of the proton conductivity to figure out the mechanism. The work is quite far away from the quality of NC publication.

Response: We thank the reviewer very much for the kind comments and suggestions.

The main finding of our manuscript is that the interactions between protons and the layer frameworks in layered solid acid $H_nM_nZ_2O_{3n+5}$ are substantially reduced at the monolayer limit, which increases the concentration of active protons and consequently improves the proton conductivities of monolayer $H_nM_nZ_2O_{3n+5}$ assembled membranes significantly. We agree with the reviewer that the directional dependence of the proton conductivity might be a possible reason for the improved proton conductivity of *m*- $H_3Sb_3P_2O_{14}$ membranes compared to the pellets made by $H_3Sb_3P_2O_{14}$ particles considering that the latter ones have engineering issue and may not have a highly oriented nanochannels as *m*- $H_3Sb_3P_2O_{14}$ membranes do. Thus, as the reviewer pointed out, comparing the proton transport properties of $H_3Sb_3P_2O_{14}$ single crystals and *m*- $H_3Sb_3P_2O_{14}$ membranes is important to clarify the mechanism for the improved proton conductivity. Unfortunately, the synthesis of large single crystals of $H_nM_nZ_2O_{3n+5}$ is very challenging (Deniard-courant, S., et al, *Solid State Ion.* 27, 189, 1988) and no single crystals large enough for proton conductivity testing are available so far. Notably, all the membranes made by $H_3Sb_3P_2O_{14}$ nanosheets with four different average thicknesses (from 1.0 nm to 8.6 nm, including monolayers) have no engineering issue and they show significantly improved proton conductivity as the thickness of nanosheets reduces. Thus, to confirm the mechanism of significantly improved proton conductivity in *m*- $H_3Sb_3P_2O_{14}$ membranes, we thoroughly investigated the structure, in particular the orientation degree, of $H_3Sb_3P_2O_{14}$ membranes made by $H_3Sb_3P_2O_{14}$ nanosheets with four different average thicknesses.

As shown in Figure R12, R13, the four kinds of membranes show almost the same

XPS, Raman and FT-IR spectra, indicating almost the same chemical composition, bonding and crystal structure of the nanosheets with different average thicknesses. Moreover, these nanosheets have similar average lateral size from 0.92 to 1.27 μm (Figure R1).

We then thoroughly characterized the orientation of $\text{H}_3\text{Sb}_3\text{P}_2\text{O}_{14}$ nanosheets in the membranes. In our experiments, all the membranes were fabricated by vacuum filtration, where the atmospheric pressure was used to squeeze the liquid towards the other side of the filter membranes and the nanosheets will be aligned by the directional water flow due to high aspect ratio (lateral size/thickness $>10^3$). SEM and XRD measurements clearly show that all the membranes have well-aligned layered structure (Figure R14 and R15). Wide-angle X-ray scattering (WAXS) pattern derives from the diffraction of an incident X-ray beam parallel to the surface of a membrane and has been widely used to quantitatively characterize the orientation degree of 2D nanosheets in the membranes (e.g., Wan, S., et al, *Science* 374, 96, 2021 & *Nat. Mater.* 20, 624, 2021). We quantitatively characterized the orientation degree of nanosheets in the four kinds membranes made of $\text{H}_3\text{Sb}_3\text{P}_2\text{O}_{14}$ nanosheets with different average thicknesses by using WAXS (Figure R16). The orientation degree of $\text{H}_3\text{Sb}_3\text{P}_2\text{O}_{14}$ nanosheets was quantified by using the Herman's orientations factor (f), which was defined as

$$f = \frac{3\langle \cos^2 \phi \rangle - 1}{2} \quad (1)$$

Where $\langle \cos^2 \phi \rangle$ is the average value of the square of the cosine of the azimuthal angle for the (003) peak of the membranes, which was calculated as

$$\langle \cos^2 \phi \rangle = \frac{\int_0^{\pi/2} I(\phi) \sin \phi \cos^2 \phi d\phi}{\int_0^{\pi/2} I(\phi) \sin \phi d\phi} \quad (2)$$

Where $I(\phi)$ is the peak intensity at an azimuthal angle of ϕ .

As shown in Figure R15 and R16, all the membranes show a set of high-order diffraction peak along (00 l) crystal plane and very similar f values from 0.97 to 0.99 for the (003) peak, confirming that these membranes have highly oriented structure along (00 l) crystal plane with almost the same orientation degree. However, as shown in our manuscript, these four kinds of membranes show significantly different proton

conductivities. Among them, the m - $\text{H}_3\text{Sb}_3\text{P}_2\text{O}_{14}$ membranes assembled by monolayers (~ 1.0 nm thick) show the highest proton conductivity over the investigated temperature range at 100% RH, which is about 2, 3 and 4 times larger than that of the membranes assembled from $\text{H}_3\text{Sb}_3\text{P}_2\text{O}_{14}$ nanosheets with an average thickness of ~ 1.4 , 3.1 and 8.6 nm, respectively. Moreover, the lateral size of nanosheets has negligible influence on the proton conductivity, as shown in Figure R3. Therefore, these results give strong evidence that the significantly improved proton conductivity in m - $\text{H}_3\text{Sb}_3\text{P}_2\text{O}_{14}$ membranes is attributed to the reduced thickness of the nanosheets rather than the directional dependence. Using solid state NMR, we revealed that the thickness dependence of proton transport in $\text{H}_3\text{Sb}_3\text{P}_2\text{O}_{14}$ is due to the greatly reduced interactions between protons and the layer frameworks at the monolayer limit, which results in substantial increase in the number of active protons and consequently the superhigh proton conductivities of m - $\text{H}_3\text{Sb}_3\text{P}_2\text{O}_{14}$ membranes.

We have added these data and related discussions in the revised manuscript.

Figure R12 XPS spectra of the membranes assembled by $\text{H}_3\text{Sb}_3\text{P}_2\text{O}_{14}$ nanosheets with different average thicknesses. **a-d** Survey XPS spectrum (**a**) and Sb 3d (**b**), P 2p (**c**) and

O 1s (d) XPS spectra of 1.0 nm-, 1.4 nm-, 3.1 nm-, and 8.6 nm- $\text{H}_3\text{Sb}_3\text{P}_2\text{O}_{14}$ membranes.

Figure R13 Raman (a) and FT-IR (b) spectra of 1.0 nm-, 1.4 nm-, 3.1 nm-, and 8.6 nm- $\text{H}_3\text{Sb}_3\text{P}_2\text{O}_{14}$ membranes.

Figure R14 Cross-sectional SEM images of the membranes assembled by $\text{H}_3\text{Sb}_3\text{P}_2\text{O}_{14}$ nanosheets with average thickness of ~1.0 nm (a), 1.4 nm (b), 3.1 nm (c), and 8.6 nm (d).

Figure R15 The XRD patterns of membranes assembled by $\text{H}_3\text{Sb}_3\text{P}_2\text{O}_{14}$ nanosheets with different average thicknesses, showing highly oriented structure along (00 l) crystal planes.

Figure R16 The azimuthal scan profiles for the (003) peak of 1.0 nm- $\text{H}_3\text{Sb}_3\text{P}_2\text{O}_{14}$ membranes (a), 1.4 nm- $\text{H}_3\text{Sb}_3\text{P}_2\text{O}_{14}$ membranes (b), 3.1 nm- $\text{H}_3\text{Sb}_3\text{P}_2\text{O}_{14}$ membranes

(c), and 8.6 nm- $\text{H}_3\text{Sb}_3\text{P}_2\text{O}_{14}$ membranes (d). The insets are the corresponding WAXS patterns for an incident Cu- $\text{K}\alpha$ X-ray beam parallel to the sheet plane. The derived Herman's orientation factors (f) for the (003) crystal plane of 1.0 nm-, 1.4 nm-, 3.1 nm-, and 8.6 nm- $\text{H}_3\text{Sb}_3\text{P}_2\text{O}_{14}$ membranes are ~ 0.987 , 0.966, 0.984, and 0.978, respectively, demonstrating that all the membranes have almost the same orientation degree along the (00 l) crystal plane.

REVIEWERS' COMMENTS

Reviewer #1 (Remarks to the Author):

I am happy to recommend the acceptance of this version since the authors have well addressed the comments raised by the reviewers.

Reviewer #2 (Remarks to the Author):

Thank you for answering the Reviewers' questions. The authors answered clearly that the proposed material is inadequate for fuel cell applications due to the anisotropy. Instead, the authors added data for 2D solid-state micro-supercapacitors using the proposed materials. If the materials can be used with high performance in micro-supercapacitor applications, they may be publishable in Nature Communications. Unfortunately, this reviewer does not have enough knowledge to judge the micro-supercapacitor performance. This manuscript should be reviewed by experts in this area. In order to change direction to micro-supercapacitors, the authors should resubmit the manuscript after changing the title, introduction, and discussion. It should be clear that the advancement of these materials in the device performance compared with the state-of-the-art.

Reviewer #3 (Remarks to the Author):

The manuscript has been improved. However, this reviewer still cannot be fully convinced for the publication in NC. If other two reviewers are very positive, this reviewer is alright for the editor to make the decision of the acceptance though the work is too much engineering instead of science oriented.

RESPONSE TO REVIEWERS' COMMENTS

Reviewer #1 (Remarks to the Author):

I am happy to recommend the acceptance of this version since the authors have well addressed the comments raised by the reviewers.

Response: We thank the reviewer very much for the positive comments.

Reviewer #2 (Remarks to the Author):

Thank you for answering the Reviewers' questions. The authors answered clearly that the proposed material is inadequate for fuel cell applications due to the anisotropy. Instead, the authors added data for 2D solid-state micro-supercapacitors using the proposed materials. If the materials can be used with high performance in micro-supercapacitor applications, they may be publishable in Nature Communications. Unfortunately, this reviewer does not have enough knowledge to judge the micro-supercapacitor performance. This manuscript should be reviewed by experts in this area. In order to change direction to micro-supercapacitors, the authors should resubmit the manuscript after changing the title, introduction, and discussion. It should be clear that the advancement of these materials in the device performance compared with the state-of-the-art.

Response: We thank the reviewer very much for the positive comments and valuable suggestions. As shown in Figure R1, the energy densities with the corresponding power densities of our solid-state m-HSbP₂O₈-MXene MSCs are higher than or comparable to those of the reported typical solid-state MXene-based MSCs [*Energy Environ. Sci.* 9, 2847, 2016; *Natl. Sci. Rev.* 10, nwac271, 2023; *Nano Energy* 45, 266, 2018], confirming the great potential of m-HSbP₂O₈ membranes for the applications in MSCs.

According to the editors' suggestion, we have moved the section that describes the MSC devices, and discussions of it to the Supplementary Information. We have also added the above information in the revised Supplementary Information.

Figure R1. Ragone plots of our solid-state m -HSbP₂O₈-MXene MSCs and the reported typical solid-state MXene-based MSCs^[1-3].

Reviewer #3 (Remarks to the Author):

The manuscript has been improved. However, this reviewer still cannot be fully convinced for the publication in NC. If other two reviewers are very positive, this reviewer is alright for the editor to make the decision of the acceptance though the work is too much engineering instead of science oriented.

Response: We thank the reviewer very much for the positive comments.

The main finding of our work is that the interactions between protons and the layer frameworks in layered solid acid $H_nM_nZ_2O_{3n+5}$ are substantially reduced at the monolayer limit, which significantly increases the concentration of active protons and consequently improves the proton conductivity of membranes dramatically. Based on this scientific finding, we used m -HSbP₂O₈ membrane to fabricate 2D solid-state MSCs to demonstrate its practical applications and further improve the impact of our work according to Reviewer #2's suggestion. Thus, our work is composed of both scientific discoveries and technological applications.